# A Case-Control Study on Factors of HPV Vaccination for Mother and Daughter in China

**DOI:** 10.3390/vaccines11050976

**Published:** 2023-05-12

**Authors:** Linyi Chen, Xihong Sun, Jing Luo, Yuanshan Zhang, Yu Ha, Xiaoxia Xu, Liandi Tao, Xuefeng Mu, Shengnan Gao, Yongchao Han, Chi Wang, Fuliang Wang, Juan Wang, Bingying Yang, Xiaoyan Guo, Yajie Yu, Xian Ma, Lijian Liu, Wenmin Ma, Pengmin Xie, Chao Wang, Guoxing Li, Qingbin Lu, Fuqiang Cui

**Affiliations:** 1Department of Epidemiology and Biostatistics, School of Public Health, Peking University, Beijing 100191, China; chenlinyi137@pku.edu.cn; 2Jining Center for Disease Control and Prevention, Jining 272209, China; 3Suzhou City Center for Disease Control and Prevention, Suzhou 234000, China; 4Lingcheng Center for Disease Control and Prevention, Dezhou 253500, China; 5Disease Control and Prevention Center of Jiuzhaigou County, Aba 623400, China; 6Longxi Center for Disease Control and Prevention, Dingxi 748000, China; 7Chengguan Center for Disease Control and Prevention, Lanzhou 730020, China; 8Yilan Center for Disease Control and Prevention, Harbin 154800, China; 9Nangang Center for Disease Control and Prevention, Harbin 150006, China; 10Qingfeng County Center for Disease Control and Prevention, Puyang 457000, China; 11Xiaoshan Center for Disease Control and Prevention, Hangzhou 311201, China; 12Si County Center for Disease Control and Prevention, Suzhou 234300, China; 13Jinxiang Center for Disease Control and Prevention, Jining 272200, China; 14Department of Laboratorial Science and Technology & Vaccine Research Center, School of Public Health, Peking University, Beijing 100191, China; 15Puyang Center for Disease Control and Prevention, Puyang 457000, China

**Keywords:** HPV vaccine, vaccination, health belief model, vaccination willingness

## Abstract

(1) Background: To explore the influencing factors of human papillomavirus (HPV) vaccination among mothers and daughters so as to provide evidence and strategies for improving the HPV vaccination rate of 9–18-years-old girls. (2) A questionnaire survey was conducted among the mothers of 9–18-year-old girls from June to August 2022. The participants were divided into the mother and daughter vaccinated group (M1D1), the mother-only vaccinated group (M1D0), and the unvaccinated group (M0D0). Univariate tests, the logistic regression model, and the Health Belief Model (HBM) were employed to explore the influencing factors. (3) Results: A total of 3004 valid questionnaires were collected. According to the regions, Totally 102, 204, and 408 mothers and daughters were selected from the M1D1, M1D0, and M0D0 groups, respectively. The mother having given her daughter sex education (OR = 3.64; 95%CI 1.70, 7.80), the mother’s high perception of disease severity (OR = 1.79; 95%CI 1.02, 3.17), and the mother’s high level of trust in formal information (OR = 2.18; 95%CI 1.26, 3.78) were all protective factors for both the mother and her daughter’s vaccination. The mother’s rural residence (OR = 0.51; 95%CI 0.28, 0.92) was a risk factor for vaccination of both mother and daughter. The mother’s education of high school or above (OR = 2.12; 95%CI 1.06, 4.22), the mother’s high level of HPV and HPV vaccine knowledge (OR = 1.72; 95%CI 1.14, 2.58), and the mother’s high level of trust in formal information (OR = 1.72; 95%CI 1.15, 2.57) were protective factors of mother-only vaccination. The older the mother (OR = 0.95; 95%CI 0.91, 0.99) was classed as a risk factor for mother-only vaccination. “Waiting until the daughters are older to receive the 9-valent vaccine” is the main reason why the daughters of M1D0 and M0D0 are not vaccinated”. (4) Chinese mothers had a high willingness to vaccinate their daughters with the HPV vaccine. The higher education level of the mother, giving sex education to the daughter, the older ages of mothers and daughters, the mother’s high level of HPV and HPV vaccine knowledge, a high level of perception of the disease severity, and a high level of trust in formal information were promoting factors of HPV vaccination for mother and daughter, and rural residence was a risk factor to vaccination. To promote HPV vaccination in girls from 9–18 years old, communities could provide health education to rural mothers with low education levels; the government could advocate for HPV vaccination through issuing policy documents; and doctors and the CDC could popularize the optimal age for HPV vaccination to encourage mothers to vaccinate their daughters at the age of 9–14 years old.

## 1. Introduction

The human papillomavirus (HPV) is the most common sexually transmitted virus [1], with more than 200 types. HPV is mainly transmitted sexually and through damaged epithelial surfaces [2] and is divided into high-risk and low-risk types. While low-risk HPV may result in warts on or around the genitals, anus, mouth, or throat, high-risk HPV can lead to cancer, including cervical cancer, vulvar cancer, vaginal cancer, etc. High-risk HPV infection is a main risk factor of the maintenance and progression of cervical intraepithelial neoplasia (CIN) from low grade (CIN1) to high grade (CIN2+). Low-grade cervical intraepithelial neoplasia (LSIL/CIN1) is typically self-limited with spontaneous regression, with a low risk of progression to high-grade intraepithelial lesion [3]. The HPV infection rate among women of 18 years old or older in China is about 20%, and HPV-16, 58, and 52 were the types with the highest infection rates [4,5,6]. The age group of ≤20 years and 61–65 years were two peaks of HPV infection [5]. Cervical cancer has a great disease burden, with a crude morbidity rate in China and the world of both 15.6 cases per 100,000 people. The age-standardized morbidity rate in China is 10.7 cases per 100,000 people, which is lower than the world average level. The mortality rate of cervical cancer is about 5 cases per 100,000 people in China, which is lower than the world average of 7.3 cases per 100,000 people [7]. Due to China’s huge population, China’s new cases of cervical cancer account for 11.7% of the total new cases in the world [8].

HPV vaccines can significantly reduce the burden of HPV-infection-related diseases [9]. The WHO recommends girls aged 9–14 years as the primary target groups for HPV vaccination, who are not yet sexually active [10]. In China, women from 9–45 years old are suitable for HPV vaccination, and girls 9–15 years old are regarded as the key population. There are five HPV vaccines approved in China: Cervarix, Gardasil, Gardasil 9, Cecolin, and Walrinvax. All vaccines except Gardasil require 3 doses for women aged 15 years or older and 2 doses for girls of 9–14 years, while Gardasil requires three doses for 14-year-old girls or older and two doses for girls of 9–13 years. However, before August 2022 (the investigation date), Gardasil 9 was only approved for women aged 16–26 years old, and Cervarix, Gardasil, and Gardasil 9 needed three doses for full vaccination.

HPV vaccination coverage in China was low among the vaccines of non-national immunization program. The accumulative vaccination coverage was 0.30%, 0.97%, and 2.24% in 2018, 2019, and 2020 in China, respectively, with developed provinces such as Beijing, Shanghai, and Zhejiang having higher coverages of 4.68–8.28%, and less developed provinces such as Tibet, Qinghai, and Xinjiang having lower coverages of 0.06% to 0.46% [11].

At present, most of the literature is confined to a hospital or a city, with limitations on extrapolation. There is also a lack of research focusing on primary vaccination groups (girls aged 9–14 years old). Parents’ education level, having heard about the vaccine, and their attitude towards the vaccine and awareness about cervical cancer or the HPV vaccine were associated with HPV vaccination among 14–18 year old girls [12]. Over 50% of teenage girls in Ethiopia had a favorable attitude toward the HPV vaccine [13]. Guardians make the majority of health-care decisions for girls, and mothers have a significant influence on the vaccinations of their daughters [14]. For example, when mothers have received HPV vaccines, their daughters will have a higher willingness to receive the vaccine [15]. Therefore, it is necessary to explore the influencing factors of HPV vaccination among mothers and daughters so as to provide evidence and strategies for promoting HPV vaccination among girls. To achieve this goal, we analyzed the distribution of demographic characteristics, the mother’s HPV and HPV vaccine knowledge, and the mother’s attitude towards the HPV vaccine between different vaccination groups, and we explored the protective factors and risk factors of HPV vaccination.

## 2. Materials and Methods

### 2.1. The Questionnaire

Based on the geographical and economic distribution, a total of 12 investigation sites were selected from seven provinces between June to August 2022, and convenience sampling was carried out among the mothers of girls from 9 to 18 years old. At each site, a primary school, a junior high school, and a senior high school were selected to distribute questionnaires, while the sample size was distributed equally in each grade. Given the low HPV vaccination coverage, mothers and daughters who were both vaccinated were directly sought and investigated until the sample size reached 15.


**The inclusion criteria for mothers were as follows:**
Daughter’s age was between 9–18 years old on the day of the survey.The mother and daughter had no serious disease or hormone treatment history.The daughter had no long-term sick leave.The mother was younger than 45 years of age when the HPV vaccine was released in 2016.The mother signed the informed consent form and agreed to comply with study procedures.



**Exclusion criteria were:**
Participants with critical information missing or logical contradictions in the questionnaire.Participants who were in the acute attack stage of a disease.


The Health Belief Model (HBM) was employed to divide questionnaire items into six dimensions, including HPV and HPV vaccine knowledge, perception of behavioral benefit, perception of behavioral barriers, perception of disease susceptibility, perception of disease severity, and trust in formal information. Specific items and point assignments are shown in Appendix A. Formal information trust refers to trust in the HPV vaccine information provided by hospitals, doctors, centers for disease control, vaccine manufacturers, and the government.

### 2.2. The Group Definition

According to the vaccination status of mothers and daughters, the participants were divided into the mother-daughter vaccination group (M1D1), the mother-only vaccination group (M1D0), and the non-vaccination group (M0D0). The M1D1 group included the population that reported “vaccinated” to the item “your HPV vaccination status” and reported “vaccinated” to the item “your daughter’s HPV vaccination status”. The participants of the other two groups were included according to these two items. Mothers in M1D0 and M0D0 were selected randomly according to their location in order to make the location distribution of the three groups proportional at 1:2:4.

### 2.3. Statistical Methods

Stata 17.0 (Stata Corp LP, College Station, TX, USA) was used for statistical analysis. For continuous variables, the normality test was performed. The normally distributed continuous variables were described with the mean and standard deviation and tested by the *t*-test, while abnormally distributed continuous variables were described with the median and interquartile range (IQR) and tested by the rank sum test. The categorical variables were described with the frequency and proportion and test by the chi-square test if all expected number ≥ 5 and the total number ≥ 40; otherwise, the Fisher exact probability test was used. We weighted the questions with different scores in each dimension so that the full score of each question was the same. The actual score of each dimension was divided by the total score of the dimension to obtain the relative score (between 0 and 1). The logistic regression model was used for multivariate analysis. Mother–daughter vaccination status was taken as the dependent variable, while demographic factors, scores of HPV and HPV vaccine knowledge, perception of behavioral benefit, perception of barriers, perception of disease susceptibility, perception of disease severity, and trust in formal information were taken as independent variables. The backward-stepwise method was used. When *p* < 0.100, the variable was included, and when *p* > 0.200, the variable was excluded. When not especially declared, *p* < 0.050 was regarded as statistically significant. Considering the different characteristics of among sites, informants were matched according to their sites.

## 3. Results

### 3.1. Sample Collection

A total of 3873 questionnaires were collected, and 3004 passed the quality exam (Appendix A). Among them, 213 (7.1%) were from Yilan, 231 (7.7%) from Nangang, 263 (8.8%) from Lingcheng, 206 (6.9%) from Qingfeng, 213 (7.1%) from Rencheng, 271 (9.0%) from Jinxiang, 252 (8.4%) from Dangshan, 345 (11.5%) from Si County, 249 (8.3%) from Xiaoshan, 249 (8.3%) from Chengguan, 188 (6.3%) from Longxi, and 324 (10.8%) from Jiuzhaigou (Appendix A).

There is no significant difference between the unqualified questionnaire and the qualified questionnaire in terms of the mother’s and daughter’s vaccination, the daughter’s age, ethnicity, family per capita monthly income, residence, and the proportion of sex education for daughters. However, compared with the mothers who passed the quality test, the mothers who failed were younger, with a median age of 40 (IQR 36, 44), while the median age of mothers who passed the test was 40 (IQR 37, 44; *p* = 0.020).

Among qualified participants, 103 (3.4%) were mother-daughter vaccinated, 355 (11.8%) were mother-only vaccinated, 2503 (83.3%) were both unvaccinated, and the remaining 43 (1.4%) were daughter-only vaccinated, which was excluded in this study. In the groups M1D1, M1D0, and M0D0, 102, 204, and 408 mothers were included in the case-control study (Appendix A). Case-control included and excluded mothers from the same location and found no statistical differences in the distribution of career, education level, monthly income per capita, residence, relatives with cancer, or sex education given to daughters.

### 3.2. Basic Demographic Characteristics

The median ages of mothers in M1D1, M1D0, and M0D0 were 43 years (IQR 41–45), 40 years (IQR 36–42), and 41 years (IQR 37–44), respectively. The median ages of the daughters in M1D1, M1D0, and M0D0 were 16 years (IQR 14–18), 14 years (IQR 12–16), and 15 years (IQR 12–17), respectively (Table 1). The mother’s age, the daughter’s age, and the childbearing age were different among the three groups. The M1D1 mothers and daughters were older than those in the M1D0 and M0D0 groups. The M1D1 mothers had an older childbearing age than the M0D0 mothers (Table 1).

In groups M1D1, M1D0, and M0D0, 44.1%, 58.8%, and 73.0% of mothers had a rural residence; 27.4%, 34.3%, and 47.8% had an educational background of junior high school; and 25.5%, 23.5%, and 17.7% had relatives with cancer (Table 2). The proportion of characteristics among M1D1, M1D0, and M0D0 are shown in Figure 1.

The mother’s occupation, education level, income per capita, residence, and whether she had given sex education to her daughter have differences among the three groups. M1D1 and M1D0 mothers had a lower proportion of farmers and higher proportions of education at senior high school level or above, >5000 Yuan monthly income per capita, a city residence, and having given sex education to their daughters than the M0D0 mothers. The M1D1 mothers also had a higher proportion of vaccine adverse events (AE) than the M0D0 mothers (Table 2).

### 3.3. HPV Vaccination Willingness

In M1D0, 96.1% of mothers were willing to get their daughters vaccinated. In M0D0, 66.7% of mothers were willing to receive the HPV vaccine themselves, and 84.3% of mothers were willing to get their daughters vaccinated. Among mothers who were willing to get their daughters vaccinated, the median expectation age for daughters to receive the vaccine was 15 years (IQR 12, 18) in group M1D0 and 17 years (IQR 14, 18) in group M0D0.

### 3.4. Scores of HBM Dimensions

The HBM section has good reliability, with a Cronbach’s α of 0.817.

For groups M1D1, M1D0, and M0D0, the median scores of HPV and HPV vaccine knowledge, perception of benefit, perception of barriers, perception of susceptibility, perception of severity, and trust in formal information were shown in Table 1. The rank sum test revealed that M1D1 and M1D0 mothers had higher levels of HPV knowledge, perception of benefit, and perception of severity than M0D0 mothers. M1D1 mothers had a higher level of perception of susceptibility than M0D0 mothers. M1D1 mothers had a higher level of trust in formal information than M1D0 and M0D0 mothers, and M1D0 mothers were higher than M0D0 mothers. Three groups had no significant differences in the score of perception of susceptibility (Figure 2).

The relative scores of each dimension were then divided into a high-level group and a low-level group, in order to be included in the multivariate analysis.

### 3.5. Multivariate Analysis

Multivariate analysis was performed using the logistic regression model, and the backward-stepwise method was used to select independent variables into the model.

(1)Comparison between M1D1 and M0D0: Compared with the M0D0 group, mothers who had given sex education to daughters (OR = 3.64; 95%CI 1.70, 7.80), had a high level of perception of the severity (OR = 1.79; 95%CI 1.02, 3.17), and a high level of trust in formal information (OR = 2.18; 95%CI 1.26, 3.78) had a higher probability of mother-daughter vaccination, while mothers in rural residences (OR = 0.51; 95%CI 0.28, 0.92) had a lower vaccination for themselves and their daughters (Figure 3A).(2)Comparison between M1D0 and M0D0: Compared with the M0D0 group, mothers with education levels of high school or above (OR = 2.12; 95%CI 1.06, 4.22), a high level of HPV knowledge (OR = 1.72; 95%CI 1.14, 2.58), and a high level of trust in formal information (OR = 1.72; 95%CI 1.15, 2.57) had a higher probability of mother-only vaccination, while mothers with an older age (OR = 0.95; 95%CI 0.91, 0.99) had a lower probability of mother-only vaccination (Figure 3B).(3)Comparison between M1D1 and M1D0: Compared with the M1D0 group, mothers who had given sex education to daughters (OR = 2.74; 95%CI 1.15, 6.54) and who were older (OR = 1.16; 95%CI 1.07, 1.26), had an older daughter (OR = 1.24; 95%CI 1.10, 1.41) had a higher probability of mother-daughter vaccination (Figure 3C).

### 3.6. Reasons for Not Being Vaccinated

Reasons for not receiving vaccination were divided into vaccine hesitancy and non-hesitancy. Reasons of vaccine hesitancy were further divided into three categories according to the Vaccine Hesitancy Matrix, which were (1) vaccine/vaccination-specific issues, (2) individual and group influences, and (3) contextual influences [16] (Appendix A).

Regarding the reasons of mothers for not getting the HPV vaccine, we analyzed 408 mothers in the M0D0 group. The reason with the highest frequency was the price of the vaccine (33.6%). Being older (23.0%), not knowing where to get vaccinated (21.1%), and vaccine shortage (17.6%) were the next three highest reasons (Figure 4A). The most frequently selected classification was vaccine/vaccination-specific issues (15.8%), followed by non-vaccine hesitancy (15.5%), contextual influences (10.6%), and, finally, individual and group influences (8.3%).

In total, 204 and 408 mothers in the M1D0 and M0D0 groups reported their reasons for not vaccinating their daughters with the HPV vaccine. For M1D0 mothers, the most frequently chosen reason was waiting for their daughters to be old enough to receive the 9-valent vaccine (50.0%), followed by not knowing if their daughters could be vaccinated (16.2%), vaccine shortage (14.7%), and their daughters not being sexually active yet (14.2%) (Figure 4B). The most frequently selected classification was individual and group influences (9.8%), followed by non-vaccine hesitancy (7.6%), vaccine/vaccination-specific issues (7.5%), and contextual influences (4.1%). For M0D0 mothers, the most frequently chosen reason was waiting for their daughters to be old enough to receive the 9-valent vaccine (29.7%), followed by waiting for their daughters to grow up and make the decision for themselves (24.3%), their daughters being too young (21.6%), and not knowing if their daughters can be vaccinated (18.9%) (Figure 4B). The most frequently selected classification was individual and group influences (11.2%), followed by vaccine/vaccination-specific issues (10.0%), non-vaccine hesitancy (6.7%), and contextual influences (6.0%).

**Figure 3 vaccines-11-00976-f003:**
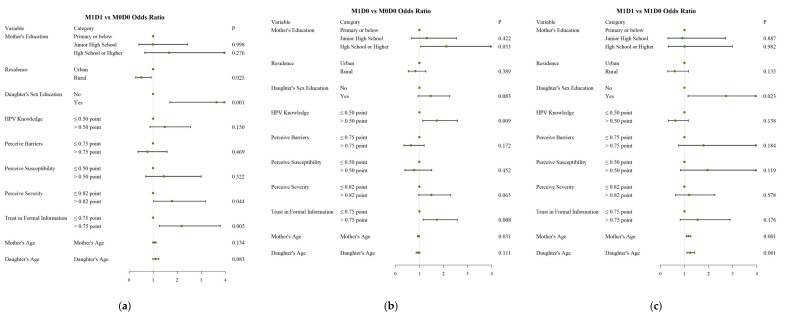
Forest plot of mother and daughter’s HPV vaccination behaviors. (**a**) M1D1 vs. M0D0; (**b**) M1D0 vs. M0D0; (**c**) M1D1 vs. M1D0.

## 4. Discussion

### 4.1. Mothers Had a High Willingness to Vaccinate Their Daughters against HPV

This study found that 96.1% and 84.3% of mothers in M1D0 and M0D0 were willing to get their daughters vaccinated, respectively, indicating a high intention. A study showed that 90.2% of guardians of junior high school students in Guangzhou were willing to vaccinate their daughters with the HPV vaccine. The high willingness of Chinese mothers to vaccinate girls aged 9–18 years against HPV is in great contrast with the low vaccination rate. A survey of four provinces of China showed that the HPV vaccination rate of girls aged 9–14 was 1.37% and that of their mothers was 2.57% [17]. China has not yet included the HPV vaccine in the national immunization program, but from 2020 to 2022, HPV vaccination has been piloted in many places, such as Inner Mongolia, Sichuan, Jiangsu, and Hainan, which aimed at female students aged 13–14 years. In order to narrow the gap between the vaccine rate and willingness, the focus should be on solving the problems encountered in actual vaccination, such as shortage of vaccine supply, the high price of the vaccine, etc.

Even though much of the focus is on vaccination among girls before the start of sexual behavior, HPV vaccines are also beneficial for patients already infected, which could ensure an earlier clearance of HPV infection in patients with low-grade cytological abnormalities [18] as well as lower the risk of developing cervical dysplasia persistence/recurrence among high-grade cervical dysplasia patients [19]. The importance of cervical screening should also not be left behind. The present cervical screening program of China was basically aimed at women between 35 and 64 years old, with 3–5 years between each screening. Screening should be carried out regardless of HPV vaccination status [20].

### 4.2. HPV Vaccination between Mother and Daughter Is Related to Demographic Characteristics and Cognition of Each Dimension of HBM

#### 4.2.1. Rural Residence

The probability of mother and daughter vaccination was lower in rural populations. There are possible explanations for this. First, the rural population is less likely to know about the HPV vaccine and has a lower understanding of HPV [21]. Second, health services are less accessible and medical workers are fewer in rural areas, making it difficult to have sufficient healthcare workers to recommend HPV vaccines. Third, the high cost of HPV vaccines may make them unaffordable for low-income rural households.

#### 4.2.2. Mother’s Education Level

Mothers with a high school level education or above had a higher probability of receiving vaccination. The possible reason is that people with a higher educational background have a greater understanding of HPV [22,23], a higher level of understanding of their own health-related issues, and more accurate health beliefs and knowledge, and will therefore make more positive health-related choices [24]. However, studies showed that HPV vaccination rates decreased with parental education when the parental education was above high school level [25,26]. This may be because parents with undergraduate or postgraduate degrees are more likely to obtain vaccine information through the Internet and other forms of media, and thus have a greater risk of obtaining inaccurate information and increasing concerns about HPV vaccines. In terms of our study, a large proportion of mothers had education of primary school and junior high school levels, and health education should be enhanced for them in order to increase their awareness of the benefits of HPV vaccines and the susceptibility of women to HPV, which could increase the probability of them vaccinating their daughters. Krawczyk’s study did not reveal a significant correlation between parents’ education and vaccination among Quebec families, which may due to the free vaccination and a history of high vaccine uptake there [27].

#### 4.2.3. Mothers Having Given Their Daughters Sex Education

Compared with the M0D0 and M1D0 groups, mothers having given sex education to their daughters was the protective factor for mother-daughter vaccination, which is in accordance with the study of Zhang et al. [28] Imparting sex education to daughters indicates that mothers had good health knowledge and attached importance to sex education for daughters. They may also have a lower sense of taboo about sex and methods to prevent sexually transmitted diseases.

#### 4.2.4. Mother’s Age

Compared with the M0D0 group, the older the mother, the lower the possibility of mother-only vaccination. Older mothers may approach the upper limit of the age range for vaccination, making it difficult to follow through with timely vaccination. Mothers who were eligible for coverage from 2016–2022, i.e., under 51 years of age, were included in the study, but mothers may be older than the recommended age because of having never heard of the HPV vaccine, having physical issues, or vaccine shortage.

When comparing with the M1D0 group, the older the mother was, the higher the possibility of mother-daughter vaccination. Naoum’s study showed older parents had higher possibilities of vaccinating their daughters [29]. This may because older mothers have higher knowledge [30] and willingness to vaccinate their daughters [31]. However, the role of a parent’s age remains controversial. Some studies have found that parental willingness to vaccinate their daughters decreased with the increasing age of the parent [32,33,34]. The reason for this is that younger parents might be more open to changing sexual norms and have more access to information. Older parents also may have relative negative beliefs towards general vaccines due to their previous experiences [32]. A study in Poland found that parents’ age did not affect their attitude towards vaccinating their children with HPV vaccines [35].

#### 4.2.5. Daughter’s Age

In our study, the older daughter was the protective factor for mother-daughter vaccination, which was in accordance with Zhang’s research [36]. It is also common for a daughter not to be vaccinated against HPV due to her age. A study found that parents considered the pre-adolescent age too young and preferred to vaccinate their daughters at the age of 16–18, probably because parents with older daughters perceived a higher risk of HPV infection [37]. However, Rancic’s study showed that children under 15 years of age were significantly more vaccinated than those ≥15 years [38]. The possible reason may be parents of children older than 15 were not well informed about HPV infection and vaccines.

The study of Yankey et al. showed that the most common reason parents did not plan to vaccinate their teenage children against HPV was because they thought their daughters were less likely to start sexual behavior, and vaccination was thus unnecessary [39]. Our study also revealed a similar phenomenon. In fact, HPV vaccines should be administered before sexual activity begins; girls under 14 years of age have not yet started sexual activity, and would achieve higher antibody levels after immunization, making it the ideal age for HPV vaccination. In September 2022, the appropriate age for 9-valent vaccination against HPV was expanded from 16 to 26 years to 9 to 45 years, which may promote HPV vaccination for girls under the age of 16.

### 4.3. Effects of HBM on HPV Vaccination Behavior

#### 4.3.1. HPV and the HPV Vaccine Knowledge

The high HPV vaccine knowledge of mothers was a protective factor for mother-only vaccination compared with the M0D0 group. High proportions of mothers in the M0D0 group did not receive the vaccination because they did not know where to go for it or had never heard of it, and high proportions of mothers in the M1D0 and M0D0 groups did not know whether or not their daughters could receive vaccination. This indicated a lack of vaccination knowledge among the unvaccinated population. A number of studies have found that a higher level of understanding of the HPV vaccine has a significant correlation with vaccination [40,41,42]. Knowledge is a key factor affecting attitudes and practices toward health behaviors; therefore, improving HPV knowledge is an important link in promoting vaccination. However, a study found that a low level of knowledge of HPV was not significantly associated with acceptance in Indonesian parents, and, therefore, special attention to existing beliefs and attitudes towards the vaccines should be paid [33].

#### 4.3.2. Perception of the Severity of the Disease

A higher level of perception of disease severity on the part of the mother was the protective factor for mother-daughter vaccination, and it was the protective factor for mother-only vaccination at the 90% confidence level. The percentages of mothers who chose “Infection with HPV usually has no serious consequences” and “My risk of HPV infection is low” as the reasons for not receiving the vaccination were 2% and 3.7%, respectively. While the proportion of participants choosing “My daughter’s risk of HPV infection is low” and “Infection with HPV usually has no serious consequences” as the reasons for their non-vaccination of daughters was less than 1%, indicating that most participants perceived the disease severity.

#### 4.3.3. Trust in Formal Information

Mothers who had a high degree of trust in formal information had a higher probability of receiving the HPV vaccine for themselves as well as their daughters. Mothers who believed in doctors, vaccine manufacturers, and government sources of information may have a higher awareness of the benefits of HPV vaccination and a high acceptance of vaccine-related services. Previous studies have shown that people who trusted more formal information had higher self-efficacy and were therefore more likely to adopt positive health behaviors [43,44]. Informal information containing personal experiences and erroneous information can be particularly difficult to correct, further hindering the promotion of accurate health information on social media [45]. Information dissemination ability and credibility of centers for disease control, the government, and doctors should be enhanced to improve mothers’ trust in formal information. At the same time, increasing the volume of voices and disseminating objective and scientific information on social media would encourage mothers to vaccinate their daughters against HPV.

### 4.4. Mothers and Daughters Did Not Receive the HPV Vaccine Predominantly Due to Non-Vaccine Hesitation Factors and the Daughter’s Age

High proportions of mothers’ non-vaccination status were due to non-vaccine hesitancy factors, including being older, a shortage of HPV vaccine supply, illness, or physical reasons. Education aimed at young mothers about HPV should be carried out as early as possible to reduce the phenomenon of untimely vaccination due to older age for uptake. The problem of vaccine shortage should be addressed by improving vaccine availability and introducing more varieties of vaccines. A total of 8.0 to 21.1% of mothers in the M0D0 group did not receive the HPV vaccine due to worries about the vaccine’s safety, doubts about the vaccine’s effect, not knowing where to get the vaccine, and having never heard about it. For these reasons, health education could be used to publicize knowledge and improve awareness.

The main reasons for daughters not getting the vaccination were age factors, including waiting for daughters to be old enough to receive the 9-valent vaccine, daughters not yet being sexually active, daughters being too young, and waiting for daughters to grow up and make the decision for themselves. This indicates a low level of awareness among mothers about the optimal age for HPV vaccination. The M0D0 mothers tended to let their daughters decide whether or not to vaccinate when they grow up, which might be related to their insufficient understanding of the HPV vaccine. In total, 50.0% and 29.7% of the mothers in the M1D0 group and the M0D0 group wanted their daughters to receive the 9-valent vaccine after they grew up, showing the mothers’ preference for the 9-valent vaccine, which was consistent with the results of Lin et al. [23]. In our study, 13.7% of mothers in the M1D0 and M0D0 groups worried about the safety of vaccinating their daughters, and 3.4% and 4.4% of mothers doubted the efficacy, which is different than Yun’s study that found that 41.8% and 31.7% parents worried about the safety and efficacy of the vaccines [46]. This difference may due to the location difference as well as China’s extended publicity campaign regarding HPV vaccines in 2022.

### 4.5. Limitations

The limitations of this study are that although regions and age were considered when including the participants, cluster random sampling was not adopted when selecting the schools. Thus, the participants may not be representative of the general population in China, and the extrapolation of the conclusions should be cautious. In this study, vaccinations before the investigation date were collected, and were not restricted to the most recent vaccination, so there may be a prevalence-incidence bias. The results of this study can also only reveal the correlation between vaccination and perception of vaccination barriers, vaccination benefits, disease severity, disease susceptibility, and trust in formal information, and cannot be identified with the causal inference. In addition, the seriousness of the participants affects the quality of the data, and only 77.6% of respondents passed the quality control questions and logic checks, so it is difficult to eliminate information bias. Due to those reasons, the results should be interpreted cautiously.

However, our study stuck to the strict research plan and applied a number of quality control links, including rigorous inclusion of informants, in-time verification of the questionnaire, arrangement of the data, and assessment of internal questionnaire consistency in order to ensure study quality. The survey was conducted in 12 sites and covered mothers with 9–18-year-old daughters, which added to the representation of the population. This study provides the most up-to-date information about mothers’ and daughters’ HPV vaccination intention and its influencing factors.

## 5. Conclusions

Chinese mothers have a high willingness to vaccinate their daughters with the HPV vaccine. Higher education levels of mothers, providing sex education to daughters, older mothers and daughters, mothers’ high knowledge level of HPV and the HPV vaccine, mothers’ high level of perception of the disease severity, and mothers’ high level of trust of formal information are all protecting factors of HPV vaccination for mothers and daughters, and rural residence is a risk factor to vaccination. In order to narrow the gap between high vaccination willingness and the low vaccination rate, the government should focus on solving the problems of vaccine supply shortage and the high cost. Health education for mothers should emphasize the optimal age for receiving the vaccine so as to reduce the expected vaccination age of mothers for their daughters.

## Figures and Tables

**Figure 1 vaccines-11-00976-f001:**
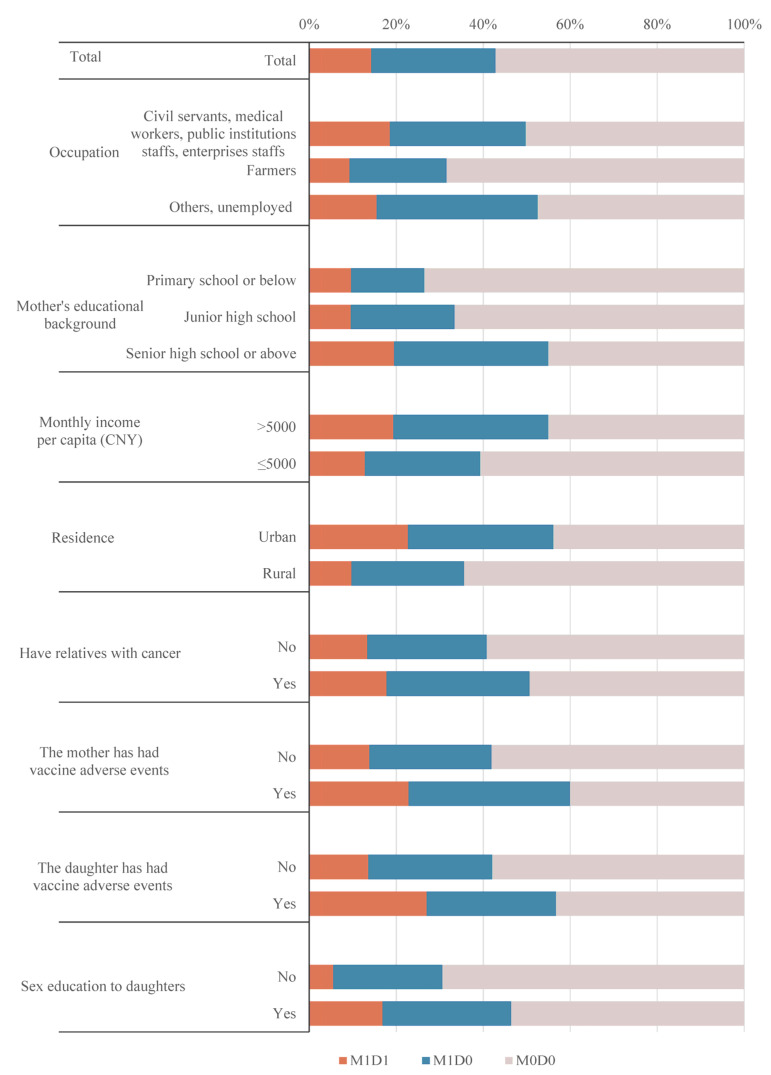
Distribution of demographic characteristics of mothers and daughters in M1D1, M1D0, and M0D0.

**Figure 2 vaccines-11-00976-f002:**
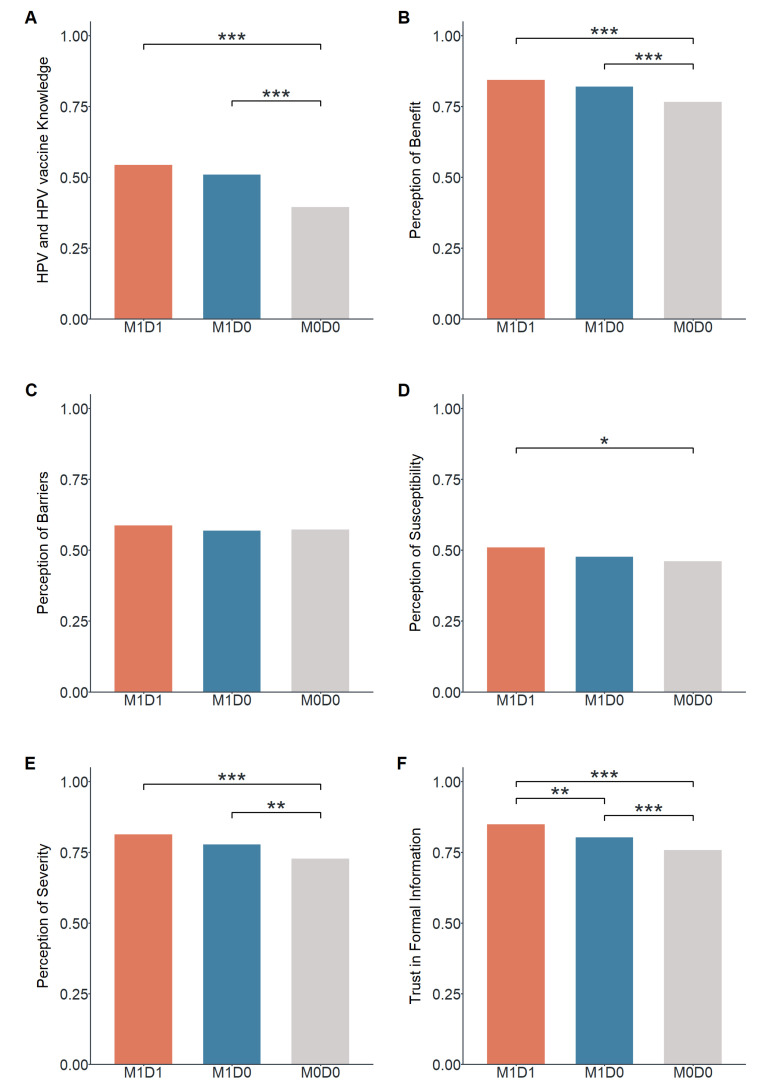
Mean scores of mothers of different HBM dimensions and comparison between groups by rank sum test (Note: * 0.010 ≤ *p* < 0.050; ** 0.001 ≤ *p* < 0.010; *** *p* < 0.001). (**A**) HPV and HPV vaccine Knowledge. (**B**) Perception of Benefit. (**C**) Perception of Barriers. (**D**) Perception of Susceptibility. (**E**) Perception of Severity. (**F**) Trust in Formal Information.

**Figure 4 vaccines-11-00976-f004:**
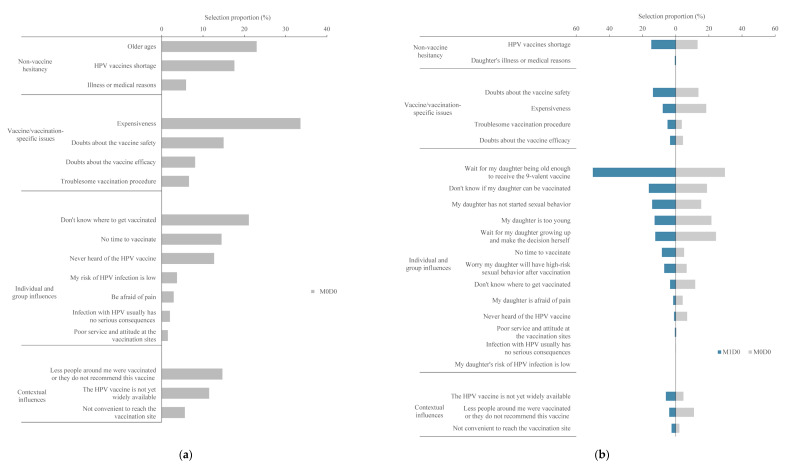
Frequency of reasons for not receiving HPV vaccine. (**a**) Mother’s reasons for not receiving HPV vaccine; (**b**) Daughter’s reasons for not receiving HPV vaccine.

**Table 1 vaccines-11-00976-t001:** Rank Sum Test of Continuous Variables for Different HPV Vaccination Behaviors of mother and daughter.

Variable	Total Median (IQR)	M1D1 Median (IQR)	M1D0 Median (IQR)	M0D0 Median (IQR)	Total *p*
Mother’s age	41	43	40	41	<0.001
	(37–43)	(41–45) ***	(36–42) ***	(37–44)	
Daughter’s age	15	16	14	15	<0.001
	(12–17)	(14–18) ***	(12–16) ***	(12–17)	
Childbearing age	25	26	25	25	0.006
	(23–28)	(25–29) *	(23–27)	(23–29)	
HPV & HPV vaccine knowledge	0.375	0.500	0.500	0.375	<0.001
	(0.250–0.625)	(0.375–0.750) ***	(0.375–0.625) ***	(0.250–0.500)	
Perception of Benefit	0.750	0.833	0.833	0.750	<0.001
	(0.750–1.000)	(0.750–1.000) ***	(0.750–1.000) ***	(0.667–0.917)	
Perception of Barriers	0.542	0.667	0.542	0.542	0.097
	(0.500–0.708)	(0.500–0.667) *	(0.500–0.667)	(0.500–0.708)	
Perception of Susceptibility	0.417	0.500	0.417	0.417	0.065
	(0.333–0.667)	(0.333–0.667) *	(0.333–0.667)	(0.333–0.583)	
Perception of Severity	0.813	0.813	0.813	0.750	<0.001
	(0.625–0.938)	(0.688–1.000) ***	(0.688–1.000) **	(0.625–0.813)	
Trust in Formal Information	0.750	0.875	0.750	0.750	<0.001
	(0.750–1.000)	(0.750–1.000) ***	(0.750–1.000) ***	(0.688–0.813)	

Note: asterisk indicates the significance of the difference between the group of the column and the M0D0 group. * 0.010 ≤ *p* < 0.050; ** 0.001 ≤ *p* < 0.010; *** *p* < 0.001.

**Table 2 vaccines-11-00976-t002:** Chi-Square Test/Fisher’s Exact Test of Categorical Variables among Mothers and Daughters of Different Vaccination Behavior.

Variables	Categories	Total	M1D1 *n* (%)	M1D0 *n* (%)	M0D0 *n* (%)	χ^2^ (*p*)
		714 (100)	102 (14.3)	204 (28.6)	408 (57.1)	
Occupation	Civil servants, medical workers, public institution staffs, enterprise staffs	307 (43.0)	57 (55.9) ***	96 (47.1) ***	154 (37.8)	<0.001
	Farmers	291 (40.8)	27 (26.5)	65 (31.9)	199 (48.8)	
	Others, unemployed	116 (16.3)	18 (17.7)	43 (21.1)	55 (13.5)	
Mother’s educational background	Primary school or below	83 (11.6)	8 (7.8) ***	14 (6.9) ***	61 (15.0)	<0.001
	Junior high school	293 (41.0)	28 (27.4)	70 (34.3)	195 (47.8)	
	Senior high school or above	338 (47.3)	66 (64.7)	120 (58.8)	152 (37.3)	
Monthly income per capita (CNY)	>5000	160 (22.4)	31 (30.4) **	57 (27.9) **	72 (17.7)	0.002
	≤5000	554 (77.6)	71 (69.6)	147 (72.1)	336 (82.4)	
Residence	Urban	251 (35.1)	57 (55.9) ***	84 (41.2) ***	110 (27.0)	<0.001
	Rural	463 (64.8)	45 (44.1)	120 (58.8)	298 (73.0)	
Have relatives with cancer	No	568 (79.5)	76 (74.5)	156 (76.5)	336 (82.4)	0.093
	Yes	146 (20.4)	26 (25.5)	48 (23.5)	72 (17.7)	
The mother has had vaccine adverse events	No	679 (95.1)	94 (92.2) *	191 (93.6)	394 (96.6)	0.094
	Yes	35 (4.9)	8 (7.8)	13 (6.4)	14 (3.4)	
The daughter has had vaccine adverse events	No	677 (94.8)	92 (90.2) *	193 (94.6)	392 (96.1)	0.056
	Yes	37 (5.2)	10 (9.8)	11 (5.4)	16 (3.9)	
Sex education for daughters	No	163 (22.8)	9 (8.8) ***	41 (20.1) *	113 (27.7)	<0.001
	Yes	551 (77.2)	93 (91.2)	163 (79.9)	295 (72.3)	

Note: asterisk indicates the significance of the difference between the group of the column and the M0D0 group. * 0.010 ≤ *p* < 0.050; ** 0.001 ≤ *p* < 0.010; *** *p* < 0.001.

## Data Availability

The data presented in this study are available on request from the corresponding author. The data are not publicly available.

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
