# Peer review of "A Case-Control Study on Factors of HPV Vaccination for Mother and Daughter in China"

_vaccines, 2023, doi:10.3390/vaccines11050976_

Round 1
Reviewer 1 Report
In my opinion, the analyzed topic is interesting enough to attract the readers’ attention. The goal of this article was to explore the influencing factors of human papillomavirus vaccination among mothers and daughters so as to provide evidence and strategies for improving the HPV vaccination rate of 9-18-years-old girls. I think that the abstract of this article is well organized and clear.
In my opinion, the discussion could be studied in depth and extended. Maybe, it could be useful the evaluation of the state of the art of preventing protocols and screening protocols in order to sensitize also young women. In particular I suggest these two articles PMID: 36992282 and PMID: 35455328. Because of these reasons, the article should be revised and completed. Considered all these points, I think it could be of interest for the readers and, in my opinion, it deserves the priority to be published after minor revisions.
The quality of English language is accettable. Typos errors sholud be corrected.
Author Response
Point 1: In my opinion, the discussion could be studied in depth and extended. Maybe, it could be useful the evaluation of the state of the art of preventing protocols and screening protocols in order to sensitize also young women. In particular I suggest these two articles PMID: 36992282 and PMID: 35455328
Response 1: Thank you for your valuable advice. We have revised the discussion by comparing the differences between our study and other studies. The latest preventing protocols were added in the introduction in line 91-96, and screening protocols were added in the discussion in line 314-317. We also cited the articles of PMID: 35455328 in the discussion on vaccine benefits in line 313-314.
Reviewer 2 Report
Dear authors,
Thank you very much for giving me the opportunity to revise the above-mentioned manuscript entitled “A Case-Control Study on Factors of HPV Vaccination for mother and daughter in China” by Linyi Chen et al.
Despite the limitations and possible mis-perfections of the study, my impression is that the particular well written and presented study should be considered for publication in your prestigious journal.
Additionally, similar publications by other scientific groups in the particular field should also be encouraged.
I do have some comments for the authors:
Line 72 Please add a sentence regarding the mild dyskariosis
Line 74 In which age groups?
Line 96 please write some sentences regarding the attitudes of young girls and the possible factors could influence their position regarding the vaccine
Discussion
Please add a paragraph presenting additional potential benefits from vaccination; explaining the effectiveness of the vaccine and the possible positive effect in already HPV positive women in terms of clearance of the disease.
(Valasoulis G, Pouliakis A, Michail G, Kottaridi C, Spathis A, Kyrgiou M, Paraskevaidis E, Daponte A. Alterations of HPV-Related Biomarkers after Prophylactic HPV Vaccination. A Prospective Pilot Observational Study in Greek Women. Cancers (Basel). 2020 May 5;12(5):1164. doi: 10.3390/cancers12051164. PMID: 32380733; PMCID: PMC7281708.)
Dear authors,
Thank you very much for giving me the opportunity to revise the above-mentioned manuscript entitled “A Case-Control Study on Factors of HPV Vaccination for mother and daughter in China” by Linyi Chen et al.
Despite the limitations and possible mis-perfections of the study, my impression is that the particular well written and presented study should be considered for publication in your prestigious journal.
Additionally, similar publications by other scientific groups in the particular field should also be encouraged.
Author Response
Point 1: Line 72 Please add a sentence regarding the mild dyskariosis
Response 1: Thank you for your advice. Since mild dyskaryosis is comparable with low-grade squamous intraepithelial neoplasia (LSIL) in the Bethesda classification, we have added description of LSIL in line 75-77. We believe your advice will help reader have a clearer grasp of HPV infection.
Point 2: Line 74 In which age groups?
Response 2: Thank you for your suggestion. The 20% infection rate was investigated among women of 18 years old or older, and we have added the description of age groups in line 77. We also added the age of the peaks of HPV infection in line 79-80.
Point 3: Line 96 please write some sentences regarding the attitudes of young girls and the possible factors could influence their position regarding the vaccine
Response 3: Thank you for your great suggestion. We have added the attitude and influence factor of girls' vaccination in line 104-107. Namely, over 50% girls were willing to receive the vaccine. Parents' education level, having heared about the vaccine, attitude towards the vaccine and awareness about cervical cancer or HPV vaccine were associated with HPV vaccina-tion among girls of 14-18 years old
Point 4: Discussion
Please add a paragraph presenting additional potential benefits from vaccination; explaining the effectiveness of the vaccine and the possible positive effect in already HPV positive women in terms of clearance of the disease.
(Valasoulis G, Pouliakis A, Michail G, Kottaridi C, Spathis A, Kyrgiou M, Paraskevaidis E, Daponte A. Alterations of HPV-Related Biomarkers after Prophylactic HPV Vaccination. A Prospective Pilot Observational Study in Greek Women. Cancers (Basel). 2020 May 5;12(5):1164. doi: 10.3390/cancers12051164. PMID: 32380733.)
Response 4: Thank you for your constructive suggestion. We have added a paragraph of vaccination benefits for HPV positive women. We found this study (PMID: 32380733) was helpful for strengthening our article. It is cited in the discussion in line 312-314.
Reviewer 3 Report
I was invited to revise the paper entitled "A Case-Control Study on Factors of HPV Vaccination for mother and daughter in China". It was a cross-sectional study aimed to evaluate attitudes towards HPV vaccination among chinese women. The study enrolled both mothers and daughters to evaluate also the influence of mother attitudes on daughter's choise. In particular, Authors divided the enrolled patients in three different groups: mother-daughter vaccination group, the mother-only vaccination group, and the non-vaccination group.
Observations:
The paper is interesting and focused on a relevant topic for public health. For my knowledge, this is one of first papers developed ion this way.
- Statistical analysis should be deeply described. It is unknown the type of analysis performed in figures 2. In addition, Authors should report the software used to perform the analysis;
- In discussion section, Authors should highlight differences with similar studies performed in other countries;
- Strenght and limitation section should be improved in discussion section;
- Authors should describe the vaccination schedule proposed in China;
Author Response
Point 1: Statistical analysis should be deeply described. It is unknown the type of analysis performed in figures 2. In addition, Authors should report the software used to perform the analysis;
Response 1: Thank you for your advice. We agree with your opinion that the statistical methods need clearer description. We have revised the method and added more details, including the software (line 164), the selection of chi-square test and Fisher exact probability test, the rule of backward-stepwise selection and the rule of matching (line165-185).
Point 2: In discussion section, Authors should highlight differences with similar studies performed in other countries
Response 2: Thank you for your suggestion. We regard it very valuable for improving our article. We have added the discussion about the differences of the correlation between vaccination and mother's education, mother's age, daughter's age and HPV knowledge. We also simplified some discussions in "rural residence" and "mothers having given their daughters sex education"
Point 3: Strength and limitation section should be improved in discussion section
Response 3: Thank you for your suggestion. We have added the limitation of unable to identify the causal inference in line 461-464. We also added a paragraph focusing on the strength, namely the strict execution, the representitve of survey sites and the latest result in line 468-474.
Point 4: Authors should describe the vaccination schedule proposed in China
Response 4: Thank you for your suggestion. We have added the vaccination schedule proposed in China in line 89-96. Basically, there are five HPV vaccines and girls under 14 or 15 years need two doses for full vaccination, and older women need three doses.
Round 2
Reviewer 3 Report
Authors properly addressed all comments